# Group Geometrical Axioms for Magic States of Quantum Computing

**Michel Planat [1,\*], Raymond Aschheim [2], Marcelo M. Amaral [2] and Klee Irwin [2]**

[1] Institut FEMTO-ST CNRS UMR 6174, Université de Bourgogne/Franche-Comté,
15 B Avenue des Montboucons, F-25044 Besançon, France

[2] Quantum Gravity Research, Los Angeles, CA 90290, USA; raymond@quantumgravityresearch.org (R.A.);
Marcelo@quantumgravityresearch.org (M.M.A.); Klee@quantumgravityresearch.org (K.I.)

[\*] Correspondence: michel.planat@femto-st.fr

**Abstract:** Let $H$ be a nontrivial subgroup of index $d$ of a free group $G$ and $N$ be the normal closure of $H$ in $G$. The coset organization in a subgroup $H$ of $G$ provides a group $P$ of permutation gates whose common eigenstates are either stabilizer states of the Pauli group or magic states for universal quantum computing. A subset of magic states consists of states associated to minimal informationally complete measurements, called MIC states. It is shown that, in most cases, the existence of a MIC state entails the two conditions (i) $N = G$ and (ii) no geometry (a triple of cosets cannot produce equal pairwise stabilizer subgroups) or that these conditions are both not satisfied. Our claim is verified by defining the low dimensional MIC states from subgroups of the fundamental group $G = \pi_1(M)$ of some manifolds encountered in our recent papers, e.g., the 3-manifolds attached to the trefoil knot and the figure-eight knot, and the 4-manifolds defined by 0-surgery of them. Exceptions to the aforementioned rule are classified in terms of geometric contextuality (which occurs when cosets on a line of the geometry do not all mutually commute).

**Keywords:** quantum computing; free group theory; Coxeter-Todd algorithm; magic states; informationally complete quantum measurementds; 3- and 4-manifolds; finite geometries

## 1. Introduction

Interpreting quantum theory is a long-standing effort, and no single approach can exhaust all facets of this fascinating subject. Quantum information owes much to the concept of a (generalized) Pauli group for understanding quantum observables, their commutation, entanglement, contextuality, and many other aspects, e.g., quantum computing. Quite recently, it has been shown that quantum commutation relies on some finite geometries such as generalized polygons and polar spaces [1]. Finite geometries connect to the classification of simple groups as understood by prominent researchers as Jacques Tits, Cohen Thas, and many others [2,3].

In the Atlas of finite group representations [4], one starts with a free group, $G$, with relations, then the finite group under investigation, $P$, is the permutation representation of the cosets of a subgroup of finite index $d$ of $G$ (obtained thanks to the Todd–Coxeter algorithm). As a way of illustrating this topic, one can refer to ([3], Table 3) to observe that a certain subgroup of index 15 of the symplectic group $S_4'(2)$ corresponds to the $2QB$ (two-qubit) commutation of the 15 observables, in terms of the generalized quadrangle of order two, denoted $GQ(2,2)$ (alias the doily). For $3QB$, a subgroup of index 63 in the symplectic group $S_6(2)$ is sufficient, and the commutation relies on the symplectic polar space $W_5(2)$ ([3], Table 7). An alternative way to approach $3QB$ commutation is in terms of the generalized

hexagon $GH(2,2)$ (or its dual), which occurs from a subgroup of index 63 in the unitary group $U_3(3)$ ([3], Table 8). Similar geometries can be defined for multiple qudits (instead of qubits).

The straightforward relationship of quantum commutation to the appropriate symmetries and finite groups was made possible thanks to techniques developed by the first author (and coauthors) that we briefly summarize. This will also be useful at a later stage of the paper with the topic of magic state quantum computing (For a relation of finite groups to anyons and universal quantum computation, see, for instance, [5]).

The remainder of this introduction recalls how having the permutation group organize the cosets leads to the finite geometries of quantum commutation (in Section 1.1) and how it allows the computation of magic states of universal quantum computing (in Section 1.2). In this paper, it is shown that magic states themselves may be classified according to their coset geometry with two simple axioms (in Section 1.3).

### 1.1. Finite Geometries from Cosets

One can refer to [3,6,7] for the material of this subsection.

Let $H$ be a subgroup of index $d$ of a free group $G$ with generators and relations. A coset table over the subgroup $H$ is built by means of a Coxeter–Todd algorithm. Given the coset table, one builds a permutation group, $P$, that is the image og $G$ given by its action on the cosets of $H$. In this paper, the software Magma [8] is used to perform these operations.

One needs to define the rank $r$ of the permutation group $P$. First, one asks that the $d$-letter group $P$ acts faithfully and transitively on the set $\Omega = \{1, 2, \cdots, d\}$. The action of $P$ on a pair of distinct elements of $\Omega$ is defined as $(\alpha, \beta)^p = (\alpha^p, \beta^p)$, $p \in P$, and $\alpha \neq \beta$. The orbits of $P$ on the product set $\Omega \times \Omega$ are called orbitals. The number of orbits is called the rank $r$ of $P$ on $\Omega$. Such a rank of $P$ is at least two, and it also known that two-transitive groups may be identified to rank two permutation groups.

One selects a pair, $(\alpha, \beta) \in \Omega \times \Omega$, $\alpha \neq \beta$, and one introduces the two-point stabilizer subgroup, $P_{(\alpha,\beta)} = \{p \in P | (\alpha, \beta)^p = (\alpha, \beta)\}$. There are $1 < m \leq r$ such non-isomorphic (two-point stabilizer) subgroups of $P$. Selecting one of them with $\alpha \neq \beta$, one defines a point/line incidence geometry, $\mathcal{G}$, whose points are the elements of the set $\Omega$ and whose lines are defined by the subsets of $\Omega$ that share the same two-point stabilizer subgroup. Two lines of $\mathcal{G}$ are distinguished by their (isomorphic) stabilizers acting on distinct subsets of $\Omega$. A nontrivial geometry is obtained from $P$ as soon as the rank of the representation $\mathcal{P}$ of $P$ is $r > 2$, and at the same time, the number of non-isomorphic two-point stabilizers of $\mathcal{P}$ is $m > 2$. Further, $\mathcal{G}$ is said to be contextual (i.e., it shows geometrical contextuality) if at least one of its lines/edges is such that a set/pair of vertices is encoded by noncommuting cosets [7].

Figure 1 illustrates that the application of the two-point stabilizer subgroup approach, just described for the index 15 subgroup of the symplectic group, is $S_4'(2) = A_6$ whose finite representation is $H = \langle a, b | a^2 = b^4 = (ab)^5 = (ab^2)^5 = 1 \rangle$. The finite geometry organizing the coset representatives is the generalized quadrangle $GQ(2,2)$. The other representation is in terms of the two-qubit Pauli operators, as first found in [1,9]. It is easy to check if lines that are not passing through the coset $e$ contain some mutually not commuting cosets so that the $GQ(2,2)$ geometry is contextual. The embedded $(3 \times 3)$-grid shown in bold (the so-called Mermin square) allows a $2QB$ proof of Kochen–Specker theorem [10].

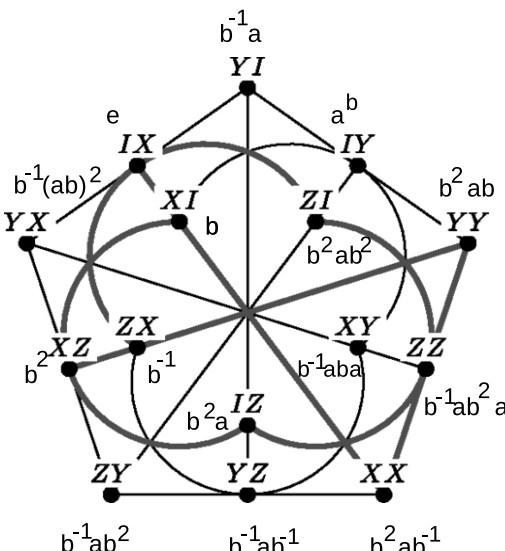

**Figure 1.** The generalized quadrangle of order two $GQ(2,2)$. The picture provides a representation of the fifteen $2QB$ observables that are commuting by triples: the lines of the geometry. Bold lines are for an embedded $3 \times 3$ grid (also called Mermin square), that is, a basic model of Kochen–Specker theorem (e.g., see ([3], Figure 1) or [10]). The second representation is in terms of the cosets of the permutation group arising from the index 15 subgroup of $G \cong A_6$ (the 6-letter alternating group).

The Kochen–Specker Theorem with a Mermin Square of Two-Qubit Observables

Let us show how to recover the geometry of the Mermin square, i.e., the $(3 \times 3)$ grid embedded in the generalized $GQ(2,2)$ of Figure 1. Recall that it is the basic model of two-qubit contextuality ([3], Figure 1; [4–7,10]). One starts with the free group $G = \langle a, b | b^2 \rangle$, and one makes use of the mathematical software Magma [8]. Then one derives the (unique) subgroup $H$ of $G$ that is of index nine and possesses a permutation representation $P$ isomorphic to the finite group $\mathbb{Z}_3^2 \times \mathbb{Z}_2^2$ reflecting the symmetry of the grid. The permutation representation is as follows;

$$P = \langle 9 | (1,2,4,8,7,3)(5,9,6), (2,5)(3,6)(4,7)(8,9) \rangle,$$

where the list $[1, ..., 9]$ means the list of coset representatives

$$[e, a, a^{-1}, a^2, ab, a^{-1}b, a^{-2}, a^3, aba].$$

The permutation representation $P$ can be seen on a torus as in Figure 2a.

Next, we apply the procedure described at the top of this subsection. There are two types of two-point stabilizer subgroups that are isomorphic either to the single element group $\mathbb{Z}_1$ or to the two-element group $\mathbb{Z}_2$. Both define the geometry of a $(3 \times 3)$ grid comprising six lines identified by their nonidentical, but isomorphic, two-point stabilizers $s_1$ to $s_6$, made explicit in the caption of Figure 2. The first grid (not shown) is considered noncontextual in the sense that the cosets on a line are commuting. The second grid, shown in Figure 2b, is contextual, in the sense that the right column does not have all its triples of cosets mutually commuting. The noncommuting cosets on this line reflect the contextuality that occurs when one takes two-qubit coordinates for the points of the grid; see [7] for more details about the relationship between noncommuting cosets and geometric contextuality.

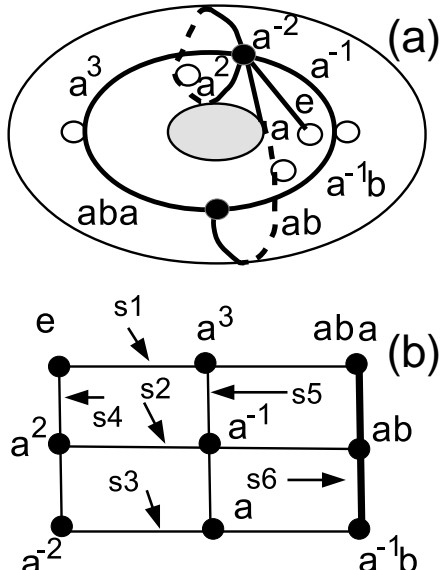

**Figure 2.** The map (**a**) leading to Mermin's square (**b**). The two-point stabilizer subgroups of the permutation representation $P$ corresponding to the dessin (one for each line) are as follows; $s_1 = (2,3)(4,7)(5,6)$, $s_2 = (1,7)(2,8)(6,9)$, $s_3 = (1,4)(3,8)(5,9)$, $s_4 = (2,6)(3,5)(8,9)$, and $s_5 = (1,9)(4,5)(6,7)$, $s_6 = (1,8)(2,7)(3,4)$, where the points of the square (resp. the edges of the dessin d'enfant) are labeled as $[1,..,9] = [e,a,a^{-1},a^2,ab,a^{-1}b,a^{-2},a^3,aba]$.

### 1.2. Magic States in Quantum Computing

Now, we recall our recent work about the relation of coset theory to the magic states of universal quantum computing. Bravyi & Kitaev introduced the principle of "magic state distillation" [11], that is, universal quantum computation (the possibility of getting an arbitrary quantum gate), which may be obtained thanks to stabilizer operations (Clifford group unitaries, preparations, and measurements) and by adding an appropriate single qubit nonstabilizer state, which is called a "magic state". Then, whatever the dimension of the Hilbert space where the quantum states live, a nonstabilizer pure state has been called a magic state [12]. An improvement of this concept was carried out in [13], showing that a magic state could correspond at the same time to a fiducial state for the construction of a minimal informationally complete positive operator-valued measure, or MIC, under the action on it of the Pauli group of the corresponding dimension. In this view, UQC is relevant both to magic states of universal quantum computation and to MICs. In [13], a $d$-dimensional magic state is obtained from the permutation group that organizes the cosets of a subgroup $H$ of index $d$ of a two-generator free group $G$.

One uses the fact that a permutation may be realized as a permutation matrix/gate and that mutually commuting matrices share eigenstates. They are either of the stabilizer type (as elements of the Pauli group) or of the magic type. One keeps magic states that are fiducial states for a MIC, because the other magic states may lead to an information loss during the computation. A catalog of the magic states relevant to UQC and MICs have been obtained by selecting $G$ as the two-letter representation of the modular group $\Gamma = PSL(2, \mathbb{Z})$ [14].

### Building a Two-Qubit MIC from a Subgroup $\Gamma_s$ of Index 4 of the Modular Group $\Gamma$

One provides an example of calculation of a MIC for the important case of two-qubit computation. The reader should refer to [13,14] for details about the concepts below. This particular case is mentioned again in Table 4 in the context of the fundamental group for the trefoil knot complement.

The permutation group of smaller size that can be used to build a four-dimensional MIC is the alternating group $A_4 = \langle v_1, v_2 \rangle$ with generators $v_1 = (1,2)(3,4) \equiv \begin{pmatrix} 0&1&0&0 \\ 1&0&0&0 \\ 0&0&0&1 \\ 0&0&1&0 \end{pmatrix}$ and $v_2 = (2,3,4) \equiv \begin{pmatrix} 1&0&0&0 \\ 0&0&1&0 \\ 0&0&0&1 \\ 0&1&0&0 \end{pmatrix}$, made explicit in terms of the (index 4) permutation representation and the corresponding permutation gate. The map for this subgroup is in Figure 3a. Using Sage [15] and the table of congruence subgroups [16], one recognizes that $\Gamma_s = \Gamma_0(3)$ whose fundamental domain is pictured in Figure 3b. In this particular case, the number of elliptic points of order two and three are $v_2 = 0$ and $v_3 = 1$, respectively; the elliptic point at $\frac{1}{2}(1 + \frac{i}{\sqrt{3}})$ is denoted by the symbol "*", and the cusps are at 0 and $\infty$. The subgroup $\Gamma_s = \Gamma_0(3)$ is generated by two transformations: $S_{\Gamma_s} : z \to \frac{z-1}{3z-2}$ and $T_{\Gamma_s} : z \to z + 1$.

The joined eigenstates of the commuting permutation matrices in $A_4$ that can serve as fiducial states for a MIC are of the form $(0, 1, -\omega_6, \omega_6 - 1) \equiv \frac{1}{\sqrt{3}}(|01\rangle - \omega_6 |10\rangle + (\omega_6 - 1) |11\rangle)$, with $\omega_6 = \exp(\frac{2i\pi}{6})$. Taking the action of the two-qubit Pauli group on the latter type of state, one finds that the corresponding pure projectors sum to four times the identity (thus form a POVM), and their pairwise distinct products satisfy the dichotomic relation $\text{tr}(\Pi_i \Pi_j)_{i \neq j} = |\langle \psi_i | \psi_j \rangle|^2_{i \neq j} \in \{\frac{1}{3}, \frac{1}{3^2}\}$. As a result, the 16 projectors $\Pi_i$ build an asymmetric informationally complete POVM (see also [13], Section 2). In Figure 3c, traces of triple products of projectors for the lines equal $\frac{1}{9}$ or $\pm\frac{1}{27}$. Instead of labeling coordinates as projectors, one labels them with the two-qubit operators acting on the fiducial state. The displayed picture corresponds to the generalized quadrangle of order two $GQ(2,2)$. It also corresponds to the set of triples of mutually commuting two-qubit operators, that is, to the picture already drawn in Figure 1. By restricting to triples of projectors whose trace is $\pm\frac{1}{27}$ one recovers the standard Mermin square at the core of Figure 1.

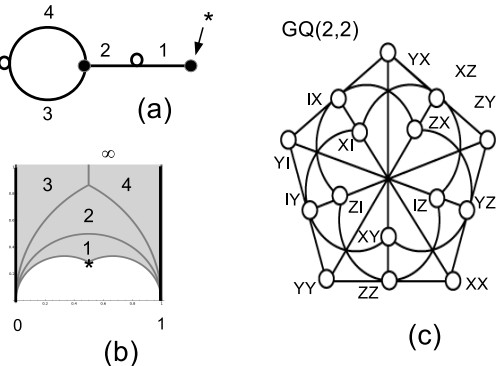

**Figure 3.** Representation of $A_4 \cong \Gamma_0(3)$ as a map (**a**) and as the tiling of the fundamental domain (the two thick vertical lines have to identified) (**b**). The character * denotes the unique elliptic point (of order 3). The organization of triple products of projectors leads to the generalized quadrangle $GQ(2,2)$ pictured in panel (**c**).

Constructing MICs Thanks to the Fundamental Group of a Knot or a Link

The next step, developed in [17,18], is to relate the choice of the starting group $G$ to three-dimensional topology. More precisely, $G$ is taken as the fundamental group $\pi_1(S^3 \setminus L)$ of a 3-manifold $M^3$ defined as the complement of a knot or link $L$ in the 3-sphere $S^3$. A branched covering of degree $d$ over the selected $M^3$ has a fundamental group corresponding to a subgroup of index $d$ of $\pi_1(M^3)$. It may be identified to a submanifold of $M^3$, the one leading to a MIC is a model of UQC. The knot involved by $\Gamma$ is the left-handed trefoil knot $T_1 = 3^1$, as shown in [14] and ([17], Section 2).

The link $L6a3$ corresponds to the congruence subgroup $\Gamma_0(3)$ of $\Gamma$, its fundamental group $\pi_1 = \langle a, b | (a, b^3) \rangle$ builds a two-qubit magic state for UQC of the type $(0, 1, -\omega_6, \omega_6 - 1)$, $\omega_6 = \exp(\frac{2i\pi}{6})$, as well as a MIC with the geometry of the generalized quadrangle of order two $GQ(2,2)$ ([17], Figure 1b). One can again refer to Table 4 below for this case. Figure 4a is the drawing of $L6a3$. There also exists a

braid representation of the link with braid word *ABCDCbaCdEdCBCDCeb* as defined in ([18], Table 1), the braid is pictured in Figure 4b.

The same two-qubit magic state and the corresponding UQC can be defined from the figure-eight knot. One can refer to Table 2 below for this case. The figure-eight manifold is hyperbolic as well as the submanifolds corresponding to the finite index subgroups of its fundamental group. Figure 4c is the drawing of *L10n46* and the corresponding braid *abCbabbcBc* is pictured in Figure 4e. The fundamental hyperbolic polyhedron is shown on Figure 4d.

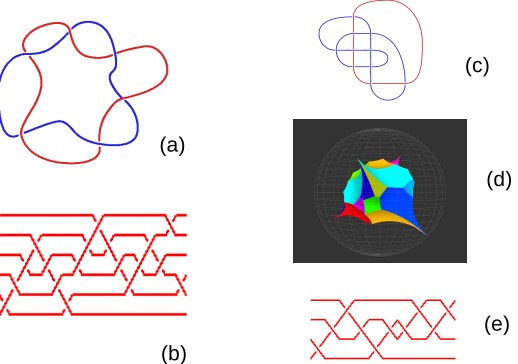

**Figure 4.** (**a**) The link L6a3 defining the two-qubit MIC from the trefoil knot and (**b**) the braid representation. (**c**) The link L10n46 defining the two-qubit MIC from the figure-eight knot, (**d**) the corresponding hyperbolic 3-manifold otet08$_{00002}$ and (**e**) the braid representation.

*1.3. Coset Geometry of Magic States*

The goal of the paper is to classify the magic states according to the coset geometry where they arise. We start from a nontrivial subgroup *H* of index *d* of a free group *G* and we denote *N* the normal closure of *H* in *G*. As above, one constructs the permutation representation *P* of *G* given by its action on the cosets of *H*. Then, seeing *P* as a group of permutation gates, the common eigenstates are either stabilizer states of the Pauli group or magic states for universal quantum computing. A subset of magic states so obtained consists of MIC states associated to minimal informationally complete measurements.

It is shown in the paper that the existence of a MIC state entails quite often that the two conditions (i) *N* = *G* and (ii) no geometry (a triple of cosets cannot produce equal pairwise stabilizer subgroups), or that these conditions are both not satisfied. In the following, we check our claim by defining the low dimensional MIC states from subgroups of the fundamental group *G* = $\pi_1(M)$ of manifolds encountered in our recent papers, e.g., the 3-manifolds attached to the trefoil knot and the figure-eight knot, and the 4-manifolds defined by 0-surgery of them.

Exceptions to the aforementioned rule are classified in terms of geometric contextuality (which occurs when cosets on a line of the geometry do not all mutually commute).

In Section 2, one deals with the case of MIC states obtained from the subgroups of the fundamental group of figure-eight knot hyperbolic manifold and its 0-surgery. In Section 3, the MIC states produced with the trefoil knot manifold and its 0-surgery are investigated.

## 2. MIC States Pertaining to the Figure-Eight Knot and Its 0-Surgery

We first investigate the relation of MIC states to the group geometrical axioms (i)–(ii) (or their negation) in the context of the figure-eight knot *K4a1* (in Section 2.2) and its 0-surgery (in Section 2.1). The fundamental group of the complement of *K4a1* in the 3-sphere *G* = $\pi_1(S^3 \setminus K4a1)$ is isomorphic to the braid group with three strands. Therefore this fundamental group is the central extension of the modular group Γ. The connection of Γ to MICs is first studied in [14] and ([17], Table 2), below are new results and corrections.

### 2.1. Group Geometrical Axioms Applied to the Fundamental Group $\pi_1(Y)$

The manifold $Y$ defined by 0-surgery on the knot $K4a1$ is of special interest as shown in ([18], Section 2) and references therein. The number of subgroups of index $d$ of the fundamental group $\pi_1(Y)$ is as follows

$$\eta_d[\pi_1(Y)] = [1, 1, 1, \mathbf{2}, 2, \ \mathbf{5}, 1, 2, \mathbf{2}, 4, \ \mathbf{3}, 17, 1, 1, 2, \ 3, 1, 6, 3, 6, \ 1, 3, 1, 43, \ \ldots],$$

where a bold number means that a MIC exists at the corresponding index.

In Table 1, one summarizes the check of our axioms (i) and (ii) applied to $\pi_1(Y)$. A triangle $\Delta$ means that a geometry does exist (corresponding to at least a triple of cosets with equal pairwise stabilizer subgroups), thus with (ii) is violated. According to our theory, for a MIC to exist, we should have (i) and (ii) satisfied, or both of them violated. The former case occurs for $d = 9$, 11 and 19. The latter case occurs for $d = 6$ where the geometry is that of the octahedron (with the 3-partite graph $K(2,2,2)$) and $d = 20$, where the geometry is encoded by the the complement of the line graph of the bipartite $K(4,5)$. In all of these five cases, a $pp$-valued MIC does exist.

For dimension 4, the bold triangle points out a violation since (i) is true and (ii) is false while the $2QB$-MIC exists. In this case the geometry is the tetrahedron (with complete graph $K_4$) but not all cosets on a line/triangle are mutually commuting, a symptom of geometric contextuality, as shown in Figure 5.

**Table 1.** Table of subgroups of the fundamental group $\pi_1[S^3 \setminus K4a1(0,1)]$ with $K4a1(0,1)$, the 0-surgery over the figure-eight knot. The permutation group $P$ organizing the cosets in column 2. If (i) is true, unless otherwise specified, the graph of cosets leading to a MIC is that of the $d$-simplex and/or the condition (ii) is true: no geometry. The symbol $\Delta$ means that (ii) fails to be satisfied. When there exists a MIC with (i) true and (ii) false, the geometry is shown in bold characters (here, this occurs in dimension 4, see Figure 5). If it exists, the MIC is $pp$-valued as given in column 4. In addition, $K(2,2,2)$ is the binary tripartite graph (alias the octahedron), and $\overline{\mathcal{L}(K(4,5))}$ means the complement of the line graph of the bipartite graph $K(4,5)$.

| d | P | (i) | pp | Geometry |
|---|---|---|---|---|
| 4 | $A_4$ | yes | 2 | 2QB MIC, $\boldsymbol{\Delta}$ |
| 5 | 10 | yes | | $\Delta$ |
| 6 | $A_4$ | no | 2 | 6-dit MIC, $K(2,2,2)$ |
| 9 | $(36,9) \cong 3^2{\rtimes}4$ | yes | 2 | 2QT MIC |
| 11 | $(55,1) = 11{\rtimes}5, (\times 2)$ | yes | 3 | 11-dit MIC |
| 16 | $(48,3) \cong 4{\rtimes}A_4$ | yes | | $\Delta$ |
| 19 | $(171,3) \cong 19{\rtimes}9$ | yes | 3 | 19-dit MIC |
| 20 | $(120,39) \cong 4{\rtimes}(5{\rtimes}(6,2))$ | yes | | $\overline{\mathcal{L}(K(4,5))}$ |

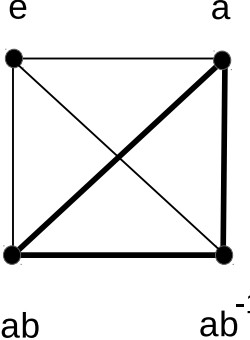

**Figure 5.** The contextual geometry associated to the 2QB-MIC and permutation group $P = A_4$ in Table 1. The line/triangle $\{a, ab, ab^{-1}\}$ is not made of mutually commuting cosets, thus geometric contextuality occurs.

## 2.2. Group Geometrical Axioms Applied to the Fundamental Group $\pi_1(S^3 \setminus K4a1)$

The submanifolds obtained from the subgroups of index $d$ of the fundamental group $\pi_1(S^3 \setminus K4a1)$ for the figure-eight knot complement are given in Table 2 (column 3) as identified in SnapPy [19].

As for the subsection above, when axioms (i) and (ii) are simultaneously satisfied (or both are not satisfied), a MIC is created; otherwise, no MIC exist in the corresponding dimension, as expected.

There are three exceptions where (i) is true and a geometry does exist (when (ii) fails to be satisfied). This first occurs in dimension 4 with a $2QB$ MIC arising from the 3-manifold otet08$_{00002}$; in this case, geometric contextuality occurs in the cosets as in Figure 5 of the previous subsection. Then, it occurs in dimension 7 (corresponding to 3-manifolds otet14$_{00002}$ and otet14$_{00035}$) when the geometry of cosets is that of the Fano plane shown in Figure 6a. Finally, it occurs in dimension 10 when the geometry of cosets is that of a $[10_3]$ configuration shown in Figure 6b ([20], p. 74). This configuration turns out to be the familiar Desargues configuration.

In addition to the latter cases, false detection of a MIC may occur (this is denoted as "fd") in dimension 8 as shown in Table 2.

**Table 2.** Table of 3-manifolds $M^3$ found from subgroups of finite index $d$ of the fundamental group $\pi_1(S^3 \setminus K4a1)$ (alias the $d$-fold coverings over the figure-eight knot 3-manifold). The covering type "ty" in column 2, the manifold identification "$M^3$" in column 3 and the number of cusps, "cp" in column 4, are from SnapPy [19]. For $d = 9$ and 10, SnapPy does not provide results so that we only identify the permutation group $P = $SmallGroup$(o, k)$ (abbreviated as $(o, k)$), where $o$ is the order and $k$ is the $k$-th group of order $o$ in the standard notation (that is used in Magma). If it exists, the MIC is "$pp$"-valued. If (i) is true, unless otherwise specified, the graph of cosets leading to a MIC is that of the $d$-simplex [and/or the condition (ii) is true: no geometry]. The symbol $\Delta$ means that (ii) fails to be satisfied. When there exists a MIC with (i) true and (ii) false, the geometry is shown in bold characters. The symbol "fd" means a false detection of a MIC when (i) and (ii) are satisfied simultaneously while a MIC does not exist. The abbreviations "Fano", "$d$-ortho", and "$[10_3]$" are for the Fano plane, the $d$-orthoplex, and the Desargues configuration.

| d | ty | $M^3$ (or $P$) | cp | (i) | pp | Geometry |
|---|---|---|---|---|---|---|
| 2 | cyc | otet04$_{00002}$, $m206$ | 1 | no | | |
| 3 | cyc | otet06$_{00003}$, $s961$ | 1 | no | | |
| 4 | irr | otet08$_{00002}$, $L10n46$, $t_{12840}$ | 2 | yes | 2 | 2QB MIC, $\Delta$ |
| | cyc | otet08$_{00007}$, $t12839$ | 1 | no | | |
| 5 | cyc | otet10$_{00019}$ | 1 | no | | |
| | irr | otet10$_{00006}$, $L8a20$ | 3 | yes | | $\Delta$ |
| | irr ($\times 2$) | otet10$_{00026}$ | 2 | yes | 1 | 5-dit MIC |
| 6 | cyc | otet12$_{00013}$ | 1 | no | | |
| | irr | otet12$_{00039}$ | 1 | no | | |
| | irr ($\times 2$) | otet12$_{00038}$ | 1 | yes | 10 | 6-dit MIC |
| | irr | otet12$_{00041}$ | 2 | no | | |
| | irr ($\times 2$) | otet12$_{00017}$ | 2 | no | | |
| | irr ($\times 4$) | otet12$_{00000}$ | 2 | yes | 2 | 6-dit MIC |
| 7 | cyc | otet14$_{00019}$ | 1 | no | | |
| | irr ($\times 4$) | otet14$_{00002}$, $L14n55217$ | 3 | yes | 2 | 7-dit MIC, $\Delta$ : *Fano* |
| | irr ($\times 4$) | otet14$_{00035}$ | 1 | yes | 2 | 7-dit MIC, $\Delta$ : Fano |
| 8 | cyc | otet16$_{00026}$ | 1 | no | | |
| | irr ($\times 2$) | otet16$_{00035}$ | 1 | no | | |
| | irr ($\times 2$) | otet16$_{00079}$ | 2 | yes | | fd |
| | irr ($\times 2$) | otet16$_{00016}$ | 2 | yes | | fd |
| | irr | otet16$_{00092}$ | 2 | no | | |
| | irr | otet16$_{00091}$ | 2 | yes | | 16-cell |
| | irr | otet16$_{00013}$, $L14n17678$ | 2 | no | | |
| 9 | | $(36, 9) \cong 3^2 \rtimes 4$ | | yes | 2 | 2QT MIC |
| | ($\times 2$) | $(504, 156) = PSL(2, 8)$ | | yes | 3 | 2QT MIC |
| | ($\times 2$) | $(216, 153) \cong 3^2 \rtimes (24, 3)$ | | yes | 2 | 2QT MIC |
| 10 | ($\times 6$) | $(160, 234) \cong 2^4 \rtimes 10$ | | yes | 5 | 10-dit MIC |
| | ($\times 2$) | $(120, 34) = S_5$ | | yes | 4 | 10-dit MIC, $\Delta$ : $[10_3]$ |
| | ($\times 2$) | $(120, 34) = S_5$ | | no | 7 | 10-dit MIC, 5-ortho |
| | | $(360, 118) = A_6$ | | yes | 5 | 10-dit MIC |

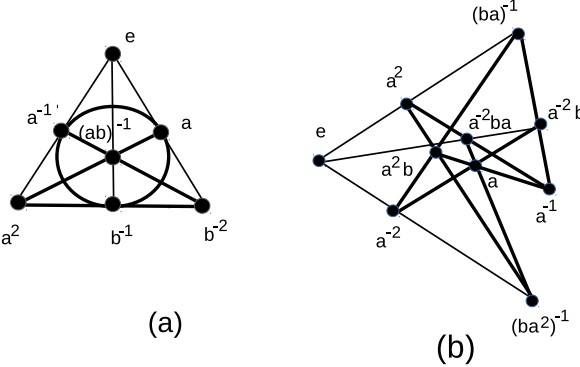

(a)

(b)

**Figure 6.** Contextual geometries associated to (i) true and (ii) false for the MICs of the figure-eight knot *K4a*1 listed in Table 1. (**a**) The Fano plane related to the manifold otet14$_{00002}$ at index 7, and (**b**) the Desargues configuration $[10_3]$ at index 10. The bold lines are for cosets that are not all mutually commuting. Each line corresponds to pair of cosets with the same stabilizer subgroup isomorphic to $\mathbb{Z}_2^2$.

## 3. MIC States Pertaining to the Trefoil Knot and Its 0-Surgery

We now investigate the relation of MIC states to the group geometrical axioms (i–ii) (or their negation) in the context of the trefoil knot $3_1$ (in Section 3.2) and its 0-sugery (in Section 3.1). The fundamental group of the complement of $3_1$ in the 3-sphere $G = \pi_1(S^3 \setminus 3_1)$, as well as its connection to MICs, is studied in [14] ([17], Table 1) and below.

### 3.1. Group Geometrical Axioms Applied to the Fundamental Group $\pi_1(\tilde{E}_8)$

The manifold $\tilde{E}_8$ is defined by 0-surgery on the trefoil knot $3_1$ and is of special interest as shown in ([18], Section 3) and references therein. The number of subgroups of index $d$ of the fundamental group $\pi_1(Y)$ is as follows.

$$\eta_d[\tilde{E}_8] = [1, 1, \mathbf{2}, \mathbf{2}, 1, \ \mathbf{5}, \mathbf{3}, 2, 4, 1, \ 1, 12, \mathbf{3}, 3, \mathbf{4}, \ 3, 1, 17, \mathbf{3}, 2, \ \mathbf{8}, 1, 1, 27, 2, \ \ldots]$$

where a bold number means that a MIC exists at the corresponding index.

Such cases are summarized in Table 3. As expected, this occurs when axioms (i) and (ii) are both true or both false. The latter case occurs at index 6 with geometry of the octahedron (and graph $K(2,2,2)$) and at index 15 with a geometry of graph $K(5,5,5)$.

**Table 3.** Table of subgroups of the fundamental group $\pi_1[S^3 \setminus 3_1(0,1)]$, with $3_1(0,1)$ the 0-surgery over the trefoil knot, when the condition (i) is satisfied or when a MIC is missed. See the captions of Tables 1 and 2 for the meaning of abbreviations.

| d | P | (i) | pp | Geometry |
|---|---|-----|-----|----------|
| 3 | 6 | yes | 1 | Hesse SIC, $\Delta$ |
| 4 | $A_4$ | yes | 2 | 2QB MIC, $\Delta$ |
| 6 | $A_4$ | no | 2 | 6-dit MIC, $K(2,2,2)$ |
| 7 | $(42,1) \cong 7 \rtimes (6,2)$ | yes | 2 | 7-dit MIC |
| 9 | $(54,5) \cong 3^2 \rtimes (18,3), (\times 2)$ | yes | | $K(3,3,3)$ |
| 12 | $(72,44) \cong 2^2 \rtimes (18,3)$ | yes | | $\mathcal{L}(K(3,4))$ |
| 13 | $(78,1) \cong 13 \rtimes (6,2), (\times 2)$ | yes | 4 | 13-dit MIC |
| 15 | $(150,6) \cong 5^2 \rtimes (6,2), (\times 2)$ | no | 6 | 15-dit MIC, $K(5,5,5)$ |
| 16 | $(96,72) \cong 2^3 \rtimes A_4$ | yes | | $K(4,4,4,4)$ |
| 19 | $(114,1) \cong 19 \rtimes (6,2)$ | yes | 3 | 19-dit MIC |
| 21 | $(126,9) \cong 7 \rtimes (18,3), (\times 2)$ | yes | 5 | 21-dit MIC, $\Delta$: $\mathbf{K(3,3,3,3,3,3)}$ |

Exceptions to the rules are when a MIC exists with (i) true but not (ii). This occurs in dimension 3 (for the Hesse SIC), as the free group has a single generator (a trivial case) at index 4 (for the 2*QB* MIC),

with a contextual geometry as in Figure 5, and at index 21, with a contextual geometry (not shown) of graph $K(3, 3, 3, 3, 3, 3, 3)$.

### 3.2. Group Geometrical Axioms Applied to the Fundamental Group $\pi_1(S^3 \setminus 3_1)$

The characteristics of the submanifolds obtained from the subgroups of index $d$ of the fundamental group $\pi_1(S^3 \setminus 3_1)$ for the trefoil knot complement are given in Table 4, using SnapPy [19] and Sage [15] for identifying the corresponding subgroup of the modular group $\Gamma$ [14] (this improves upon the method in ([17], Table 1)).

As for the above sections, when axioms (i) and (ii) are simultaneously satisfied (or both are not satisfied), a MIC is created; otherwise, no MIC exist in the corresponding dimension, as one should expect.

There are a few exceptions where (i) is true and a geometry does exist (when (ii) fails to be satisfied). This first occurs in dimension 3 for the Hesse SIC where the free subgroup is trivial with a single generator. The next exceptions are for the 6-dit MIC related to the permutation group $S_4$ with the contextual geometry of the octahedron shown in Figure 7a; in dimension 9 for the $2QT$ MIC related to the permutation group $3^3 \times S_4$, with a contextual geometry consisting of three disjoint lines; and in dimension 10 for a 10-dit MIC related to the permutation group $A_5$ and the contextual geometry of the so-called Mermin pentagram. The latter geometry is known to allow a $3QB$ proof of the Kochen–Specker theorem [10].

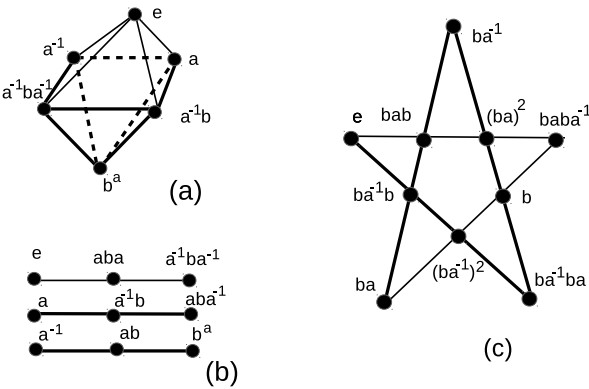

**Figure 7.** Contextual geometries associated to (i) true and (ii) false for the MICs of the trefoil knot $3_1$ listed in Table 2. (**a**) The octahedron, related to the subgroup $\Gamma_0(4)$ of $\Gamma$ at index 6; (**b**) three disjoint lines $K_3^3$ at index 9; (**c**) and the Mermin's pentagram at index 10. The bold lines are for cosets that are not all mutually commuting.

**Table 4.** Subgroups of index $d$ of the fundamental group $\pi_1(S^3 \setminus 3_1)$ (alias the $d$-fold coverings over the trefoil knot 3-manifold). The meaning of symbols is as in Table 2. When the subgroup in question is a subgroup of the modular group $\Gamma$, it is identified as a congruence subgroup or by its signature $NC(g, N, \nu_2, \nu_3, [c_i^{W_i}])$ (see [14] for the meaning of entries). The permutation group $P =$ SmallGroup$(o, k)$ is abbreviated as $(o, k)$. As in Table 1, if (i) is true, unless otherwise specified, the graph of cosets leading to a MIC is the $d$-simplex and/or the condition (ii) is true. Exceptions (with geometry identified in bold characters) are for a MIC with (i) true and (ii) false. For indices 9 and 10, some subgroups of large order could not be checked as leading to a MIC or not; they are not shown in the table. The abbreviation octa is for the octahedron, MP is for the Mermin pentagram, and $K_3^3$ means three disjoint triangles.

| d | ty | cp | P | (i) | pp | Type in $\Gamma$ | Geometry |
|---|---|---|---|---|---|---|---|
| 2 | cyc | 1 | $(2,1) \equiv 2$ | no | | | |
| 3 | cyc | 1 | $(3,1) \equiv 3$ | no | | | |
| | irr | 2 | $(6,1) \equiv 6$ | yes | 1 | $\Gamma_0(2)$ | Hesse SIC, $\mathbf{\Delta}$, L7n1 |
| 4 | cyc | 1 | $(4,1) \equiv 4$ | no | | | |
| | irr | 2 | $(12,3) = A_4$ | yes | 2 | $\Gamma_0(3)$ | 2QB MIC, $\mathbf{\Delta}$, L6a3 |
| | irr | 1 | $(24,12) = S_4$ | yes | 2 | $4A^0$ | 2QB MIC |
| 5 | cyc | 2 | $(5,1) \equiv 5$ | no | | | |
| | irr | 3 | $(60,5) = A_5$ | yes | 1 | $5A^0$ | 5-dit MIC |
| 6 | reg | 3 | $(6,1) \equiv 6$ | no | 2 | $\Gamma(2)$ | 6-dit MIC, $6_3^3$ [18] |
| | cyc | 3 | $(6,2) = 3 \times 2$ | no | | $\Gamma'$ | |
| | irr | 2 | $A_4$ | no | 2 | $3C^0$ | 6-dit MIC, K(2,2,2) |
| | irr | 1 | $(24,13) = 3 \rtimes 8$ | no | | $6B^0$ | |
| | irr | 1 | $(18,3) \cong 3^2 \rtimes 2$ | no | | $6A^0$ | |
| | irr | 3 | $S_4$ | yes | 2 | $\Gamma_0(4)$ | 6-dit MIC, $\mathbf{\Delta}$ : octa |
| | irr | 2 | $A_5$ | yes | 2 | $\Gamma_0(5)$ | 6-dit MIC |
| | irr | 2 | $S_4$ | yes | 2 | $4C^0$ | 6-dit MIC, $\mathbf{\Delta}$ : octa |
| 7 | cyc | 1 | $(7,1) \equiv 7$ | no | | | |
| | irr ($\times 2$) | 2 | $(42,1) \cong 7 \rtimes (6,2)$ | yes | 2 | $NC(0,6,1,1,[1^16^1])$ | 7-dit MIC |
| | irr ($\times 2$) | 1 | $(168,42) = PSL(2,7)$ | yes | 2 | $7A^0$ | 7-dit MIC |
| | irr ($\times 2$) | 2 | $S_7$ (order 5040) | yes | | $NC(0,10,1,1,[2^15^1])$ | 7-dit MIC |
| 8 | cyc | 1 | $(8,1) \equiv 8$ | no | | | |
| | irr | 2 | $(24,13)$ | no | | $6C^0$ | |
| | irr | 2 | $S_4$ | no | | $4D^0$ | |
| | irr ($\times 2$) | 2 | $(24,3) \cong 2.A_4$ | yes | | | 16-cell |
| | irr | 2 | $PSL(2,7)$ | yes | | $\Gamma_0(7)$ | fd |
| | irr ($\times 2$) | 1 | $SL(2,7)$ | yes | | $NC(0,8,2,2,[8^1])$ | fd |
| | irr ($\times 2$) | 2 | $(48,29) \cong 2.(24,3)$ | yes | | $8A^0$ | 16-cell |
| 9 | | | $(9,1) \equiv 9$ | no | | | |
| | | 2 | $(18,3)$ | no | 7 | $6D^0$ | 9-dit MIC, K(3,3,3) |
| | | 2 | $(54,5) \cong 3^2 \rtimes (18,3)$ | no | 7 | $NC(0,6,3,0,[3^16^1])$ | 9-dit MIC, K(3,3,3) |
| | | 1 | $(324,160) \cong 3^3 \rtimes A_4$ | no | | $9A^0$ | K(3,3,3), $K_3^3$ |
| | | 3 | $(54,5)$ | yes | 7 | $NC(0,6,1,0,[1^12^16^1])$ | 9-dit MIC |
| | ($\times 3$) | | $(162,10) \cong 3^2 \rtimes 6$ | yes | | | K(3,3,3) |
| | ($\times 2$) | 1 | $(504,156) = PSL(2,8)$ | yes | 3 | $NC(1,9,1,0,[9^1])$ | 2QT MIC |
| | ($\times 2$) | 2 | $(432,734) \cong 3^2 \rtimes (48,29)$ | yes | 2 | $NC(0,8,3,0,[8^11^1])$ | 2QT MIC |
| | | 3 | $(648,703) \cong 3^3 \rtimes S_4$ | yes | 2 | $NC(0,12,1,0,[2^13^14^1])$ | 2QT MIC, $\Delta : K_3^3$ |
| 10 | | 1 | $(120,35) \cong 2 \rtimes A_5$ | no | | $10A^0$ | 5-ortho |
| | | 2 | $A_5$ | yes | 5 | $5C^0$ | 10-dit MIC, $\Delta$: **MP** |
| | ($\times 2$) | 1 | $(720,764) \cong A_6 \rtimes 2$ | yes | 9 | $NC(0,10,0,4,[10^1])$ | 10-dit MIC |

## 4. Conclusions

Previous work on the relationship between quantum commutation and coset-generated finite geometries has been expanded here, by establishing a connection between coset-generated magic

states and coset-generated finite geometries. The magic states under question are those leading to MICs (with minimal complete quantum information in them). We found that, given an appropriate free group $G$, two axioms, (i) the normal closure $N$ of the subgroup of $G$ generating the MIC is $G$ itself and (ii) no coset-geometry should exist, or the negation of both axioms (i) and (ii), are almost enough to classify the MIC states. The few exceptions rely on configurations that admit geometric contextuality. We restricted the application of the theory to the fundamental group of the 3-manifolds defined from the figure-eight knot (an hyperbolic manifold) and from the trefoil knot, and to 4-manifolds $Y$ and $\tilde{E}_8$ obtained by 0-surgery on them. It is of importance to improve of knowledge of the magic states due to their application to quantum computing, and we intend to pursue this research in future work.

**Author Contributions:** All authors contributed substantially to the research. Conceptualization: M.P. and K.I.; methodology, M.P.; software, M.P. and R.A.; validation, R.A. and M.M.A.; formal analysis, M.P. and M.M.A.; investigation, M.P., R.A., and M.M.A.; resources, R.A and K.I.; data curation, M.P.; writing, original draft preparation, M.P.; writing, review and editing, M.P. and M.M.A.; supervision, K.I.; project administration, M.P. and K.I.; funding acquisition, K.I.

**Funding:** The research was funded by Quantum Gravity Research, Los Angeles, CA, USA.

**Conflicts of Interest:** The authors declare no conflicts of interest.

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
