# Peer review of "Group Geometrical Axioms for Magic States of Quantum Computing"

_mathematics, doi:10.3390/math7100948_

Round 1

Reviewer 1 Report

This paper continues the work of the authors. It is well-written and I recommend to publish it in the present form.

The fokus of the paper is two-fold.

At the first the construction of magic states using finite groups (related to finite geomnetries) and secondly its relation to contextuality (Kochen-Specker). The paper is well-written and will contribute to the topic with new results. But the whole paper based strongly on previous work of the authors, i.e there are many self-citations.

The whole problem is not completely new for quantum computing, see the hidden subgroup problem.

Also the realization of quantum gates by using finite groups was discussed before (e.g. quant-ph/0206128). The authors used finite groups but I miss the motivation why these groups have to be the fundamental groups of 3- and 4-manifolds. It is known that any finitely representable group is the fundamental group of a 4-manifold (see Gompf,Stipsicz: 4 manifolds and kirby calculus for a proof). Maybe the author should mention this result.

Furthermore, in subsection 3.2 the authors discussed the index d subgroups of the fundamental group of the trefoil knot. This fundamental group is isomorphic to the braid group with three strands and therefore this fundamental group is the zentral extension of the modular group (there is a beautiful proof of Milnor).

The author should also mention this fact as a motivation to for the identification of these subgroups with the subgroups of the modular group.
Finally I recommend for publication but by considering a minor revision with the points above.

Author Response

Following the opinion of the academic editor, we only performed minor changes in the paper to account for the remarks by the referees.
The minor corrections suggested by referee 1 are as follows
1. We added footnote 1 and a citation to [5] (arxiv quant-ph/0206128).
2. We mentioned again the relevance of modular group to the braid group on three strands (at the beginning of Sec. 3.1).

Reviewer 2 Report

In general, connecting quantum phenomena (magic states here) to geometry (finite geometry) is an interesting direction so, in principle, the paper is a welcomed addition to the herd. However, I have two observations/recommendations: 

1) The paper is hard to read which frustrates most enthusiastic readers. A more extended background review where all terms, notations acronyms and methods are introduced, properly explained and motivated, is an absolute necessity. I strongly recommend dedicating a descent section for that (rather than sending the reader to nine previous papers!).  

2) It is not obvious to me how to reproduce any of the results/tables of the paper or even certify them, so a sentence like "...as identified in SnapPy [15]
(this corrects a few mistakes of [13, Table 2])" won't appear again. I would like to see the codes and reproduce the results before accepting the paper. I also recommend that the codes and any relevant detail of the calculations to be added as an appendix to the paper.

Author Response

Following the opinion of the academic editor, we only performed minor changes in the paper to account for the remarks by the referees.
The minor corrections suggested by referee 1 are as follows
1. We added footnote 1 and a citation to [5] (arxiv quant-ph/0206128).
2. We mentioned again the relevance of modular group to the braid group on three strands (at the beginning of Sec. 3.1).

Concerning the revisions proposed by referee 2, we are sorry that the paper may be hard to understand. But we think that the subsections
in the introduction are sufficient to introduce the concepts needed in our paper that are geometric contextuality in Sec. 1.1 and magic
states in Sec 1.2. Our goal, announced in Sec. 1.3, seems to us simple enough: finding group axioms to relate finite geometries, finite
groups and magic states.
It would need a review paper to follow the two recommendations of this referee: (1) a self-contained background review, (2) the codes to
certify the results. The codes are not difficult to establish and are available on demand. However it would be cumbersome (not a minor)
task to provide them in an annex in order to make them useful.
Following the remark of this referee, we corrected the sentence at the beginning of Sec. 2.2.

Round 2

Reviewer 2 Report

My concerns were not addressed. I need to see them resolved. Please refer to my previous report. 

Author Response

Following the comments by the academic editor we took into account what the referee 2 was expecting.

1. Concerning the code we display it in the private response to referee 2. There are a few Magma programs that we used to produce the results in this paper.
They are not clean enough to be included in an annex of the paper.

2. Concerning an expanded introduction to our concepts, we did it. The added subsections and figures are highlighted in the pdf. Of course,
there is anoverlap with some sections of our previous papers. We agree that it should help to digest the content of this paper by a non-expert reader
and we thank referee 2 for asking us to somehow clarify the presentation.
The new subsections are

* \subsubsection*{The Kochen-Specker theorem with a Mermin square of two-qubit observables}

* \subsubsection*{Building a two-qubit MIC from a subgroup $\Gamma_s$ of index $4$ of the modular group $\Gamma$}

* \subsubsection*{Constructing MICs thanks to the fundamental group of a knot or a link}

and the Figures 2 to 4 inside.

Round 3

Reviewer 2 Report

 None